# Determinants of Evapotranspiration in Urban Rain Gardens: A Case Study with Lysimeters under Temperate Climate

**Ahmeda Assann Ouédraogo** [1,*]**, Emmanuel Berthier** [1,*]**, Brigitte Durand** [2] **and Marie-Christine Gromaire** [3]

[1] Equipe TEAM, Centre d'Etudes et d'Expertise sur les Risques, l'Environnement, la Mobilité et l'Aménagement (Cerema), 12 rue Teisserenc de Bort, F 78190 Trappes, France

[2] Division Etudes et Ingénierie, Direction de la Propreté et de l'Eau, Service Technique de l'Eau et de l'Assainissement (DPE-STEA), 27 rue du Commandeur, F 75014 Paris, France; brigitte.durand1@paris.fr

[3] Leesu, Ecole des Ponts, Université Paris Est Creteil, F 77455 Marne-la-Vallee, France; marie-christine.gromaire@enpc.fr

* Correspondence: ahmeda.ouedraogo@cerema.fr (A.A.O.); emmanuel.berthier@cerema.fr (E.B.)

**Abstract:** Accurate evaluation of evapotranspiration (ET) flux is an important issue in sustainable urban drainage systems that target not only flow rate limitations, but also aim at the restoration of natural water balances. This is especially true in context where infiltration possibilities are limited. However, its assessment suffers from insufficient understanding. In this study, ET in 1 m$^3$ pilot rain gardens were studied from eight lysimeters monitored for three years in Paris (France). Daily ET was calculated for each lysimeter based on a mass balance approach and the related uncertainties were assessed at ±0.42 to 0.58 mm. Results showed that for these lysimeters, ET is the major term in water budget (61 to 90% of the precipitations) with maximum values reaching 8–12 mm. Furthermore, the major determinants of ET are the existence or not of an internal water storage and the atmospheric factors. The vegetation type is a secondary determinant, with little difference between herbaceous and shrub configurations, maximum ET for spontaneous vegetation, and minimal values when vegetation was regularly removed. Shading of lysimeters by surroundings buildings is also important, leading to lower values. Finally, ET of lysimeters is higher than tested reference values (evaporimeter, FAO-56, and local Météo-France equations).

**Keywords:** evapotranspiration estimation; urban rain gardens; lysimeters; evapotranspiration models

## 1. Introduction

Urbanization has a great impact on cities' hydrological cycle: runoff is increased to the detriment of infiltration and evapotranspiration (ET), leading to an increase in risks linked to flooding and deterioration of the receiving environments. Urban stormwater management policies have been developed in recent years that favour runoff management in green infrastructure systems (GIS) in order to store the water before to infiltrate, evaporate and transpire it. These sustainable urban drainage systems (SUDS) are considered as a viable mechanism that can substitute or complete the traditional sewerage system (canalisation, underground basins, pipes, etc.) and also provide environmental benefits apart from hydraulic services [1–4]. SUDS uses a set of GIS, such as green roofs, rain gardens, infiltration basins, rain trees, etc.

Rain gardens are recognised as one of the best stormwater management practices in countries such as Northern Europe, the United States, Canada, Japan, and Australia, since in addition to reducing the runoff, they also allow for water treatment and promote biodiversity in the urban environment [5,6]. Rain gardens are, by definition, a local structure with a shallow depression that receives rainwater from upstream can infiltrate, evaporate, transpire, or treat this water [7,8]. Significant hydrological processes in a rain garden include the exfiltration to the underlying soil or by drainage system, the evapotranspiration

and the interception from vegetation. These processes "should work together" for being able to control large flows and reduce the total volume of small storms [9].

Jennings et al. [10], in a study on the efficiency of residential rain gardens in terms of runoff reduction, in Ohio, temperate climate in the USA, attribute a major role to the exfiltration process and a minor role to the evapotranspiration as regards their contributions of 85% and 0.32% respectively, in reducing runoff volumes. However, more recent experimental studies [11–16] have shown a greater importance of ET in GIS. In rain gardens, daily ET rates are generally low, around 1–5 mm per day, a rate that is sufficient to restore the retention capacity of the structure between two rain events [17]. Studies estimated the ET between 43 and 70% [8] and sometimes up to 78% of the collected rainfall [13].

The ET is known as a dynamic process and it depends on meteorological factors (e.g., precipitation characteristics, air relative humidity and temperature and wind speed), GIS properties (e.g., drainage system, soil, etc.), and vegetation [15]. While the ET process has been investigated widely in agriculture, it remains relatively unknown in urban areas, and particularly in SUDS. Even though progress has been made in the study of ET in urban areas, in particular with the development of approaches based on remote sensing, the current models are still imprecise and do not always account for all the specificities (spatial heterogeneity, microclimatic variability, etc.) associated with the urban environment [18–21].

In the hydrological modelling aspect of these GIS structures, the representation of the water transfer processes in the soil (infiltration of water in the soil, exfiltration, etc.) have been prioritized in the preliminary studies. A review of 11 urban hydrological models used for modelling in SUDS, including rain gardens, by Kaykhosravi et al. [19] also noted that despite recent improvements in existing models, their ability to model multi-layered soil systems, trees or vegetation processes (interception, absorption, and evapotranspiration), snowmelt, and runoff at different spatial scales is limited and further research are needed. In these hydrological models, the ET is usually estimated and represented by predictive equations based on physical approaches that require significant input data (Penman–Monteith [22] model is a reference and its variants of Fao-56 [23] or ASCE [24] methods) or other more conceptual approaches that use less data (Hargreaves and Allen [25], Priesley–Taylor [26]). These predictive equations have been evaluated with the estimated ET in pilot rain gardens lysimeters in the literature [12–14,27]. The findings of these research show that the classical equations for ET are not always satisfactory with either underestimations or overestimations of the observed ET data. Another method proposed by Hess et al. [17], and based on water content measurements at different soil depths seems to be less expensive in terms of input data, and provides comparable results to the classical assessment methods of Penman–Monteith [22] and Hargreaves and Allen [25]. The main limitation of using water content profiles can be their non-representativeness of the spatial variation in water content in gardens due to its important heterogeneity.

For urban rain gardens, recent research has shown the significance of ET, but there are not enough case studies estimating the flux and the factors involved. Note that this lack is particularly related to the difficulty in measuring the flux on the one hand and, on the other hand, the fact that some preliminary studies have minimised its importance [15]. Thus, to the challenges of stormwater management and also urban heat islands, ET in rain gardens is a topic receiving more and more attention from both rain garden designers for a better consideration of ET in the design and hydrology researchers for a more accurate description of the flux in the urban context.

In some countries, such as Australia and the United States, legislation is already taking form to include ET in the design of rain gardens [17]. In France, the Paris Council with its "ParisPluie" plan seeks to develop the rain garden method [28,29]. The city has instrumented eight rain garden lysimeters for a better understanding and prediction of their hydrological behaviour. In order to extrapolate on real situations, experimental rain gardens of reduced size and well-known structures were designed. Monitoring was carried out with lysimeters, i.e., mechanisms that enable the water balance components (exfiltration, water storage, etc.) to be observed, with measurements by weighing the variations in water

content of the lysimeter. The aim was also to test different vegetation configurations and internal storage options, and to implement replicas in order to test the validity of the measurements. In this study, the purpose consists of three main points: estimate the actual evapotranspiration (ET) of these rain gardens at daily steps; assess the impact of different configurations on ET fluxes; and compare the actual ETs obtained from the lysimeters with reference to ET values, such as evaporation, from a pan evaporimeter and some models taken from the literature.

## 2. Materials and Methods

### 2.1. General Context of the Study Area

The site is located at 43 rue Buffon in Paris, France, within the Museum National d'Histoire Naturelle (MNHN) (Figure 1). In the Paris region, there is no strong topographical contrast and the agglomeration of Paris is very dense, with an estimated population of nearly two million people and 9 million in the 1500 cities and villages that constitute its suburbs [29]. Paris has a fairly temperate climate, with moderately warm summers (average temperature of 19 °C in July) and moderately cold winters (average temperature of 3 °C in January), with rare snow. The urban dominance leads to urban heat islands (UHIs), characterised by night-time temperatures that are about 2.5 °C higher (annual average) compared to rural areas [30]. The average annual rainfall of 650 mm is evenly distributed over the year and the annual potential ET is in average around 850 mm with higher values in summer and limited values in winter (data from Météo-France, The French Meteorological Service).

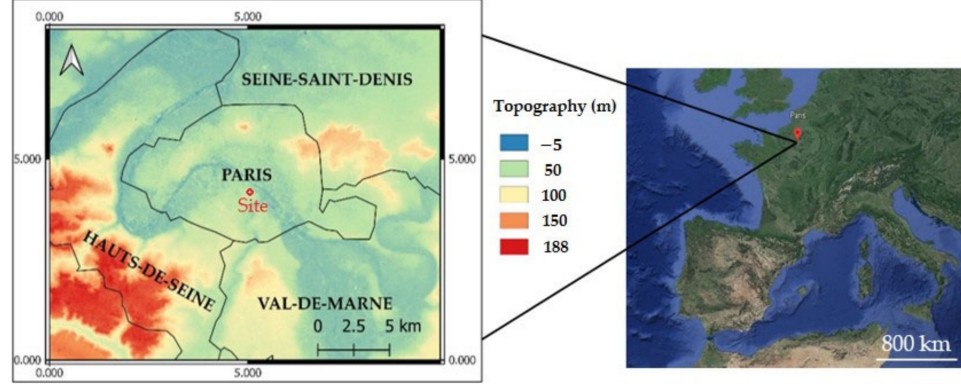

**Figure 1.** Situation of the study area (a red point) in the city of Paris (France), with the coordinate system of RGF93, Lambert 93. Topographic data source is from the site urs.earthdata.nasa.gov accessed on 24 February 2022.

### 2.2. Experimental Set Up, Data Acquisition, and Validation

#### 2.2.1. Experimental Set Up

A concrete slab of about 35 m² supports eight lysimeters, each one made up of a 1 m³ pilot rain garden (1 m × 1 m × 1 m) and a cone to increase the impluvium to 4 m² (Figure 2). Near the lysimeters, a meteorological station (Figure 2a), which consists of a pyranometer, an anemometer placed at a height of 2 m from the surface, a temperature sensor, and a hygrometer provides climatic data (global radiation, wind speed and direction, air temperature and humidity, and atmospheric pressure).

As the lysimeters are above the soil and therefore not insulated thermally compared to a situation in the ground, a 10 cm of expanded polystyrene insulation was added to all the vertical walls of lysimeters. At the bottom of each lysimeter, a 0.2 m layer of a manufactured alveolar product is installed to store rainwater (Nidaplast® product with a void index of 0.95) and a piezometer is installed to measure the water level in the internal water storage (IWS). The soil, with a thickness of 0.8 m in each lysimeter represents a natural silty-clay soil used in the city's parks and gardens of Paris region; it contains little limestone, 18 to 25%

of clay, with a neutral to basic pH (7.5–8). Weighing cells and a tipping bucket allow the measurement of mass variation and exfiltration at the bottom of each lysimeter respectively (see Figure 3). A pan evaporimeter with a diameter of 1.2 m was also installed to control the quantity of water evaporated.

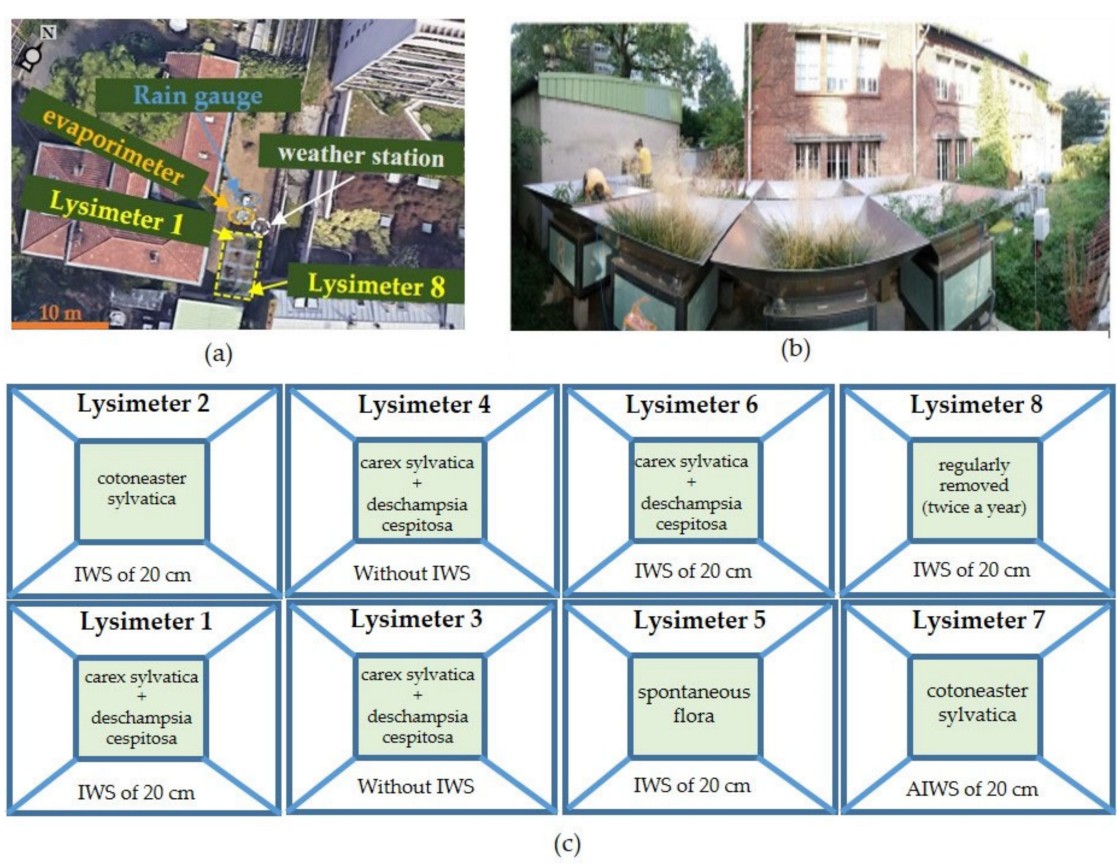

**Figure 2.** A top and panoramic views of the site in figures (**a**) (Source: google earth) and (**b**), respectively. The figure (**c**) illustrates the positions and the scientific names of vegetation in each lysimeter.

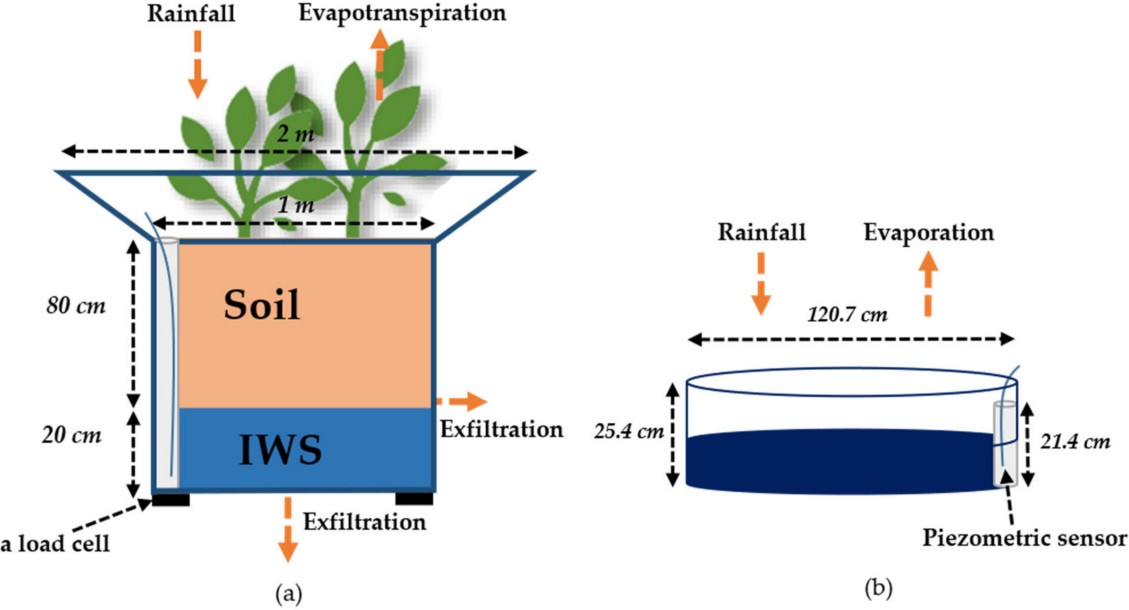

**Figure 3.** Schematical representation of water fluxes on lysimeters (**a**) and evaporimeter with an overflow of 21.4 cm (**b**). IWS refers to the internal water storage.

The configurations of the lysimeters are numbered from 1 to 8—they differ by the vegetation type and the drainage conditions (presence or not of IWS) (Figure 2c):

- The reference configuration (lysimeters 1 and 6) includes the internal water storage (IWS; i.e., the drainage at the bottom of the lysimeter, which is located just above the alveolar product), with an herbaceous stratum (6 plants of Carex sylvatica and Deschampsia cespitosa, which are native to the Paris region). This configuration is considered as the reference because of the Paris subsoil context (heterogeneous and sensitive areas of gypsum or former mines, etc.), and the importance to anticipate the impact of waterproof systems on climate change;
- Lysimeters 2 and 7 differ from the reference by a modification of the vegetation with a shrub layer (3 Cotoneaster lacteus plants per lysimeter). These plants are from China and are often used in Paris plantations;
- Lysimeters 3 and 4 differ from the reference by the lack of IWS, i.e., the water is evacuated at the bottom of the alveolar product;
- Lysimeter 8 is similar to the reference but without vegetation (spontaneous vegetation is removed twice a year);
- Lysimeter 5 is similar to the reference but with spontaneous vegetation.

### 2.2.2. Data Acquisition and Validation

The data (Tables 1 and 2) were collected at two-minute time step for a period of about 3 years (24 November 2016 to 26 December 2019). The analysis and the validation of data were carried out at daily steps. For all variables, the maintenance days were removed, whereas maintenance used to be three times a month. The variables involved in the water balance were analysed in the following way. First, for very rainy days, rainfall values were compared with the measurement from a nearby rain gauge of Météo-France (the French meteorological service, situated at 1 km); if our rain gauge data were very different from the reference data of Météo-France, they were considered as non-valid. In addition, the exfiltration data of the lysimeters with reserve were compared with the data of the water level measurement in this reserve. The idea was to have zero exfiltration when the storage is not filled (<20 cm) for lysimeters with IWS.

**Table 1.** Details of materials used for measurement on each lysimeter (the accuracy is expressed in equivalent mm of water in a lysimeter).

| Materials | Variables | Accuracy (mm) |
| --- | --- | --- |
| Bucket flow meter (PRÉCIS-MECANIQUE, 3029/2) | Cumulative exfiltration (l) | 0.008 |
| Piezometric sensor (PARATRONIC, EN61000-6-2) | Water level (mm) in the IWS | 1 mm |
| Load cells (SKAIM, FT-SK30X-FEG-0603) | Lysimeter's mass (kg) | 0.36 mm |

After removing the false, the aberrant, and the missing values, over the 1096 days that represented the three years, the percentage of validated data for the precipitation, the exfiltration and the mass variation were, respectively, 82%, 70–83%, and 66–76% (Table A1).

For the pan evaporimeter, in winter days, during rainy periods, the water level measurement (L) frequently reaches its maximum; therefore, an overflow occurs and the level variation is then set to zero. In these periods, the condition is that the water level (L) added to the rainfall should be less than the threshold of the measurement ($L_{max}$) that has been defined as equal to 170 mm; a maximum value that varied due to the fluctuation of the sensor during maintenance.

**Table 2.** Details of materials used for measuring meteorological data.

| Materials | Variables |
|---|---|
| Temperature and humidity sensor (LSI-LASTEM, DMA672) | Temperature (°C) and Air humidity (HR en %) |
| Rain gauge (LSI-LASTEM, DQA131.1) | Rain (mm) |
| Evaporimeter (Pan, LSI-LASTEM, DYI010) | Water level (mm) |
| Global radiometer iso cl-2 (LSI-LASTEM, DPA053) | Global incoming solar radiation (Watt/m$^2$) |
| Anemometer (LSI-LASTEM, DNA202) | Wind speed (m/s) |
| Barometer (LSI-LASTEM, DQA24) | Atmospheric pressure (hPa) |

*2.3. Methods*

2.3.1. Water Balance

Daily ET is calculated for each lysimeter based on the following equation:

$$ET = 4*P - Exf - \Delta S \tag{1}$$

with ET the evapotranspiration (mm), P the cumulative rainfall measured with the rain gauge (mm), Exf the cumulative exfiltration (mm), and $\Delta S$ the mass variation (mm) of the considered lysimeter.

For the daily evaporation (E, mm) from the pan evaporimeter, it is expressed as the difference between the daily cumulated rainfall (P, mm) and the daily water level variation ($\Delta L$, mm):

$$E = P - \Delta L \tag{2}$$

2.3.2. Evaluation of Measurement Uncertainty

The assessment of the uncertainties associated with the ET estimations is based on the law of the propagation of uncertainties [31]:

$$u(Y)^2 = \sum_{k=1}^{n} u(X_k)^2 \left(\frac{\partial f}{\partial X_k}\right)^2 + 2\sum_{k=1}^{n-1}\sum_{j=k+1}^{n} u(X_k, X_j)\left(\frac{\partial f}{\partial X_k}\right)\left(\frac{\partial f}{\partial X_j}\right) \tag{3}$$

where f is the function of n measured variables $X_k$, $u(X_k)$ the standard uncertainty, $u(X_k, X_j) = u(X_k)u(X_j)r(X_k, X_j)$ the estimated covariance of $X_k$ and $X_j$ with $r(X_k, X_j)$ the correlation coefficient.

By applying the Equation (3) to the balance equation (Equation (1)), it gives:

$$u(ET)^2 = \left(u(P)^2\left(\frac{\partial ET}{\partial P}\right)^2 + u(dM)^2\left(\frac{\partial ET}{\partial \Delta S}\right)^2 + u(exf)^2\left(\frac{\partial ET}{\partial exf}\right)^2\right)$$
$$+2\left(u(P, \Delta S)\left(\frac{\partial ET}{\partial P}\frac{\partial ET}{\partial \Delta S}\right) + u(P, Exf)\left(\frac{\partial ET}{\partial P}\frac{\partial ET}{\partial Exf}\right) + u(\Delta S, Exf)\left(\frac{\partial ET}{\partial \Delta S}\frac{\partial ET}{\partial Exf}\right)\right) \tag{4}$$

To solve the Equation (4), the first hypothesis is that the standard uncertainties associated with the rainfall and exfiltration measurements are at the maximum of a bucket tilt of 0.2 mm and 0.008 mm, respectively. The second assumption was to assume that the uncertainties of rainfall and exfiltration follow uniform laws, which permit their standard uncertainties to be re-estimated by $0.2/\sqrt{3}$ (0.115 mm) and $0.008/\sqrt{3}$ (0.00462 mm), respectively [31]. The standard uncertainty associated with the mass measurement for each lysimeter is 0.36 mm, a value obtained from the manufacturer [32]. The standard uncer-

tainty of the cumulative values is assessed by: $\sqrt{n}*u(ET)$ and the estimated uncertainties are given as a 95% confidence interval.

### 2.3.3. Comparison Tools

Different statistical tools are used to make comparisons between the different replicates or to compare the observed and modelled data. The non-parametric Wilcoxon rank test for paired samples was performed to compare the significance of differences between replicates and lysimeter configurations. The null hypothesis $H_0$ of this test suggests the same population for the distributions, while the alternative hypothesis $H_1$ assumes different distributions. The assumed risk $\alpha$ is taken at 5%. Simple regression models were also used to compare the observed replicas. Cumulations were also made by considering common days with valid data for lysimeters to be compared. Finally, to show the influence of the meteorological variables on ET, the partial least squares (PSL) analysis is performed. The variable important in the projection (VIP, see Appendix A for more details) that resumes the influence of each independent variable in a PSL model was used [33–35]. Indeed, a given variable will have a high importance for VIP $> 1$, a medium importance for VIP $> 0.8$, and a low importance for VIP $< 0.8$ [33,34,36].

### 2.3.4. Evapotranspiration Formulas

The predictive equations of ET tested here are summarized in Table 3. The two Penman–Monteith models applied on references vegetation (Fao-56 and Météo-France), the Penman and the Priestley–Taylor models will be compared with the estimated ETs from the lysimeters and the evaporation from the evaporimeter.

**Table 3.** Evapotranspiration (ET)'s formulations used in this study. The FAO and Météo-France formulations are two ways of setting parameters for Penman-Monteith (PM) equation.

| Name | Formulas | Hypotheses |
|---|---|---|
| Penman [37] | $ET_P = \dfrac{\Delta(Q^*-Q_G)+E_a\gamma}{L_e(\Delta+\gamma)}$ | $E_a = 0.35(e_s - e_a)(0.5 + 0.01u)$ |
| PM (FAO-56) [23] | $ET_{PM-FAO-56} = \dfrac{0.408\Delta(Q^*-Q_G)+\frac{900}{T+273}\gamma(e_s-e_a)}{\Delta+\gamma(1+0.34\ u)}$ | Well-watered vegetation with a height of 0.12 m, a surface resistance of 70 m/s, a surface emissivity of 1 and an albedo of 0.23. |
| PM (Météo-France) [38] | $ET_{PM-MF} = \dfrac{0.408\ \Delta(Q^*-Q_G)+\frac{(\gamma)(1297.8+1038.2u)(e_s-e_a)}{T+273}}{\Delta+(\gamma)(1.42+0.336u)}$ | Well-watered meadow with a surface resistance of 60 m/s, a surface emissivity of 0.95 and an albedo of 0.2. |
| Priestley and Taylor [26] | $ET_{PT} = \alpha_{PT}\dfrac{\Delta}{\Delta+\gamma}\dfrac{Q^*}{L_e}$ | Defined for saturated soils, the advection coefficient $\alpha_{PT}$ is set to 1.26 [26]. |

In these equations, terms are defined as follows: $Q^*$ is the net radiation (MJ/d), $Q_G$ the heat flux conducted in the soil (MJ/d), $L_e$ the latent heat of vaporization (KJ/kg), $e_s$ the saturation vapour pressure of air at surface temperature (KPa), $e_a$ the partial vapour pressure of atmosphere (KPa), and u is the wind speed (m.s$^{-1}$) at a reference level (2 m), $\Delta$ the slope of the saturation vapour curve, $\alpha_{PT}$ is the advection coefficient, $\gamma$ is the psychometric constant, and T refers to the temperature (°K).

## 3. Results

### 3.1. Estimated Evapotranspiration

In Figure 4, the meteorological variables measured at the site are presented. All variables are expressed as a daily average, except for exfiltration and rainfall, which are daily cumulated values. Seasonal dynamics specific to the temperate climate are observed for these variables. Global solar radiation ($R_G$) is higher in summer (up to 265 w/m$^2$) than in winter (max, 20 w/m$^2$). The net radiation (Figure 4a) assessed according to Allen et al. [21] is more significant in summer (up to 151 w/m$^2$) than in winter (max, 49.8 w/m$^2$).

Temperatures (T) reach the maximum at 34 °C in summer and are sometimes below 0 °C in winter. In contrast to the temperature, the air humidity ($H_R$, 29–96%) is higher in winter and lower in summer. The air pressure ($P_{atm}$) shows the same trend as the air humidity but less marked and varying between 980 and 1040 hPa. The wind speed (u) is between 0.1 and 1.6 m/s, higher in winter and lower in summer.

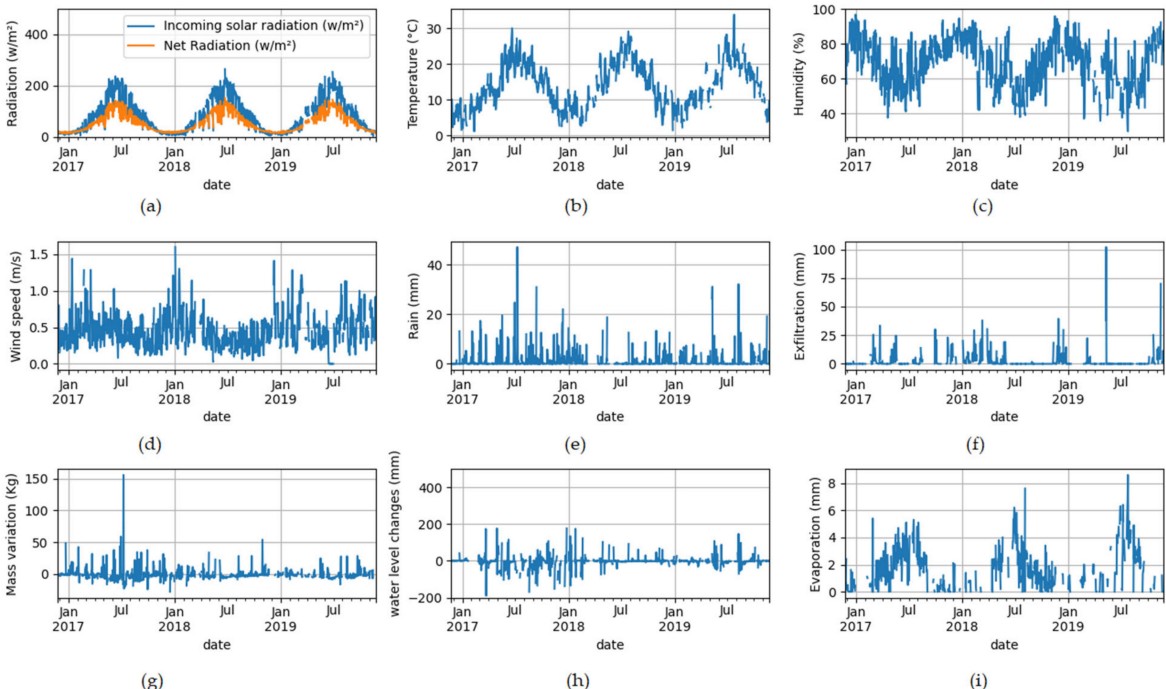

**Figure 4.** Meteorological variables: (**a**) daily incoming solar radiation and net radiation, (**b**) mean air temperature measured at 2 m, (**c**) relative air humidity, (**d**) wind speed, and (**e**) rainfall, (**f**) exfiltration, lysimeter daily (**g**) mass changes, (**h**) water level variation from the reference lysimeter 1 are added (mm), and (**i**) evaporation estimated from the evaporimeter.

For the variables specific to lysimeters (mass, water level in the IWS, and exfiltration), the reference configuration, i.e., lysimeter 1, is shown in Figure 4f–h. In addition, the Figure 4i gives the estimated evaporation from the evaporimeter with an average of 2.1 mm/d, high values in summer (max, 8.6 mm), and low values in winter.

Validated ET data after processing for the three years (1096 days) vary from 53% to 68% depending on the lysimeter (Table A1). In Figure 5, the validated ET for each lysimeter is presented. The annual dynamics of ET are shown with high daily values that can exceed 10 mm between spring and summer and small values in winter and autumn. These seasonality patterns can be linked to the atmospheric factors described above. The main atmospheric factors affecting ET in these systems are discussed later in Section 3.3.

Daily standard uncertainties and uncertainties at a 95% confidence interval are evaluated for all lysimeters (Table 4). The results uncertainties are in the range ±0.42 to ±0.58 mm for daily ET depending on the lysimeter.

**Table 4.** Associated daily ET uncertainties for each lysimeter in mm. u(ET) values refer to the standard uncertainty and 1.96 u(ET) the uncertainty for a 95% confidence interval.

|  | ET 1 | ET 2 | ET 3 | ET 4 | ET 5 | ET 6 | ET 7 | ET 8 |
|---|---|---|---|---|---|---|---|---|
| u(ET) | 0.28 | 0.24 | 0.24 | 0.21 | 0.28 | 0.29 | 0.28 | 0.23 |
| 1.96 u(ET) | 0.54 | 0.47 | 0.47 | 0.42 | 0.55 | 0.58 | 0.54 | 0.45 |

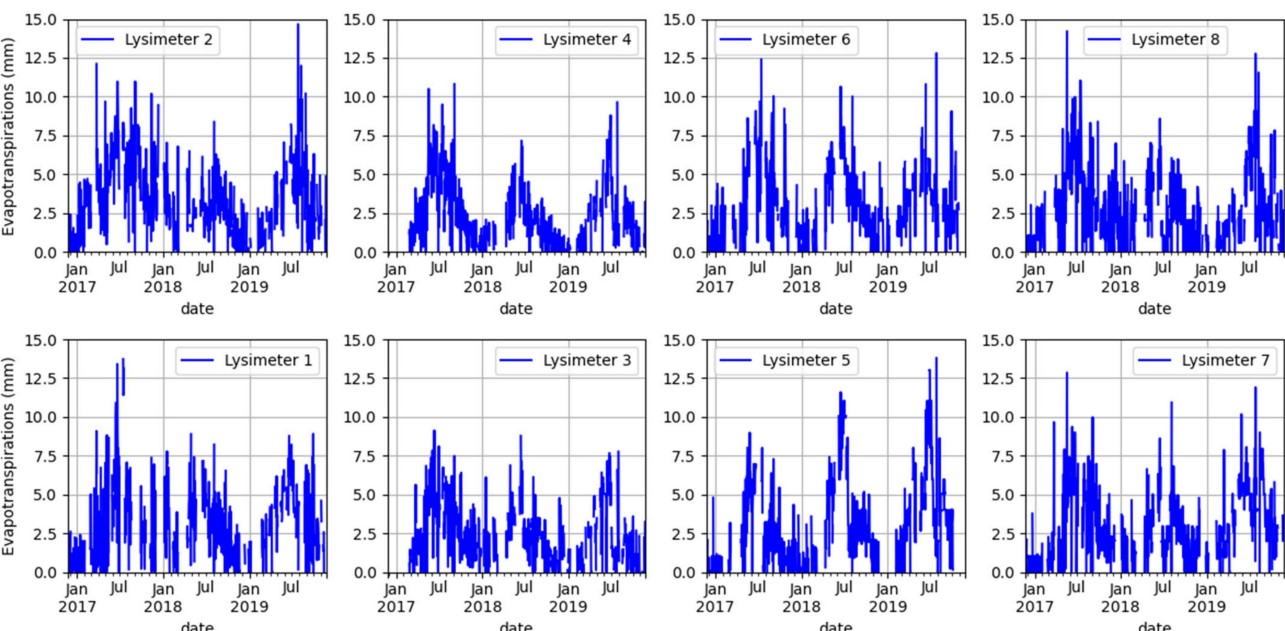

**Figure 5.** Daily evapotranspiration (ET) validated for all lysimeters.

### 3.1.1. Comparison of the Replicas

Three pairs of lysimeters (lysimeters 1 and 6, lysimeters 3 and 4, and lysimeters 2 and 7) have the same characteristics: vegetation, presence of storage or not, and the same maintenance planned during the experiment. The aim here is to compare their consistency knowing that they should be similar in term of performance. However, if a major difference is observed, this means that, for identical systems, an external variable, to be identified, is at the origin of this difference.

Based on the regression models and cumulates presented in Figure 6 for each couple of replicas, it is noted that for both lysimeters 2 and 7, if the regression model shows an acceptable fit between the two data sets $r^2 = 0.55$, ET of lysimeter 7 is clearly lower than lysimeter 2 in terms of global trend and cumulative amounts (ET2 = 1967 $\pm$ 11 mm and ET7 = 1662 $\pm$ 13 mm). Lysimeters 1 and 6 have similar trends, and cumulative amounts (ET1 = 1334 $\pm$ 11 mm, ET6 = 1330 $\pm$ 12 mm), even though the determination coefficient is low $r^2 = 0.42$ due to the underestimation and overestimation of lysimeter 6 from 01/2018 to 05/2018 and from 10/2018 to 07/2019, respectively.

For the couple lysimeters 3 and 4, it presents a coefficient of determination $r^2 = 0.57$, similar trends and a slight underestimation of lysimeter 4 in terms of cumulative data (ET3 = 1544 $\pm$ 12 mm, ET4 = 1449 $\pm$ 10 mm).

Another way of comparing these pairs is to perform statistical tests. In Table A2, results of the Wilcoxon rank test are presented. When the test is performed on the whole validated data set (3 years), only lysimeter 1 (the reference) and lysimeter 6 have similar distributions. If the test is performed by season (fall, winter, spring, and summer), different results are obtained. In all seasons, pair 1 and 6 do not show statistically different distributions, lysimeter 3 and 4 show statically different distributions in autumn and winter only while lysimeters 2 and 7 are statistically different except in spring.

It is difficult to conclude that for each replica, both lysimeters evaporated and transpired perfectly in the same way. In addition, the estimated uncertainties on ET for each lysimeter are small compared to the differences between the replicas (Figure 6a–c). However, in view of the above results, it can be said that the couples ET1/ET6 and ET3/ET4 configurations represent an acceptable replica and, the shrub configurations ET2/ET7 cannot be considered as a replica. For lysimeters 2 and 7, the only variable that differs from the two is the exposure to the buildings surrounding the installation.

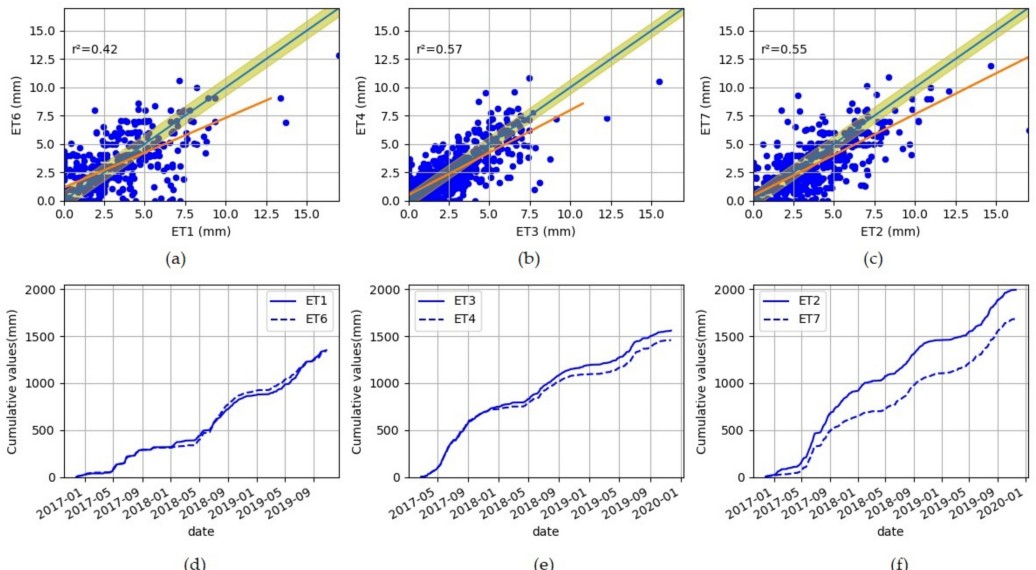

**Figure 6.** Regressions (**a**–**c**) and cumulatives (**d**–**f**) plots comparing the three replicas ET1/ET6 (data = 445), ET3/ET4 (data = 600) and ET2/ET7 (data = 583) for the three years (1096 days). The red line refers to the regression line and the blue line represents the y = x surrounded by the confidence interval (at 95%) that corresponds to the square root of the sums of the squares of the lysimeter uncertainties for each replica (0.79 mm, 0.63 mm, and 0.71 mm, respectively for ET1/ET6, ET3/ET3, and ET2/ET7).

3.1.2. Comparison between Different Configurations

The comparison of all configurations was conducted with the validated common days for the eight lysimeters from November 2016 to November 2019. These days start in mid-spring (83 days), continue throughout the summer (132 days), and end in mid-autumn (65 days). In winter, there are only 25 valid days because of the greater measurement uncertainty during the cold and rainy periods. Table 5 gives the cumulative exfiltration, mass variation, and evapotranspiration based on this common period for lysimeters.

**Table 5.** Cumulative water balance components ($C_{wb}$) in mm over the 305 common validated days, for the 8 lysimeters. Cumulated rain (4P) is 679 ± 6 mm.

| Lysimeters | | 1 | 2 | 3 | 4 | 5 | 6 | 7 | 8 |
|---|---|---|---|---|---|---|---|---|---|
| Exf | $C_{wb(Exf)}$ | 162 | 86 | 438 | 568 | 204 | 199 | 191 | 250 |
| | *% of 4P* | 24% | 13% | 65% | 84% | 30% | 29% | 28% | 37% |
| ΔS | $C_{wb(\Delta S)}$ | −543 | −486 | −492 | −633 | −675 | −577 | −471 | −427 |
| | *% of 4P* | −80% | −72% | −72% | −93% | −99% | −85% | −69% | −63% |
| ET | $C_{wb(ET)}$ | 1066 ± 7 | 1082 ± 6 | 740 ± 8 | 750 ± 7 | 1152 ± 8 | 1060 ± 7 | 962 ±8 | 864 ± 8 |
| | *% of 4P* | 157% | 159% | 109% | 110% | 170% | 156% | 142% | 127% |

The exfiltration varies between 13% (lysimeter 2) and 84% (lysimeter 4) of the input rainfall. Free drainage configurations (lysimeter 3 and 4) naturally exfiltrated the most water compared to the others set up with IWS and account for about three times (438 mm and 568 mm) of the standard configurations (lysimeter 1 and 6, 162 and 199 mm respectively). Furthermore, the herbaceous configurations (1 and 6) exfiltrated more than the shrub configurations (2 and 7); although, for lysimeters 6 and 7, this difference is reduced. Finally, the exfiltration capacity of the configuration with regularly removed vegetation (lysimeter 8, 250 mm) is higher compared to the other lysimeters with IWS and that could imply a contribution of the vegetation to the decrease in seepage.

For stock changes (ΔS), the eight settings always have negative values between −633 and −425 mm. Indeed, most of the validated common days are spring and summer days

with low rainfall, which are favourable periods for ET. Therefore, for a given day, the mass change is negative meaning that the system (lysimeter) loses water. This explains the negative cumulative $\Delta S$ observed here.

### 3.2. Determinants of ET in Lysimeters

To carry out the analysis in this section, common data of lysimeters were considered in pairs, in order to increase the number of samples and the representativeness of all seasonal periods. These numbers are noted in the text or in the Table A2. It is also important to remind that the experimental set-up was installed to test the impact of three main factors on the water balance in rain gardens. These factors are lysimeter storage (absence or presence of IWS), vegetation type and management, and local meteorological variables.

- Impact of the storage in the lysimeter structure.

Installing an IWS is globally favourable to the ET, and to the reduction in the exfiltration (Table 5). These differences are notable in all seasons. Indeed, the Wilcoxon test between lysimeter 1 and lysimeters 3 and 4 show that the distributions of estimated ET data are different in all seasons and over the whole three years (Table A2). In addition, from Table 5 or Table A7, considering the percentage of ET sum to the collected rainfall (4P), the ET of lysimeter 1 is more compared to the other two lysimeters. Compared to lysimeters 3 and 4, in autumn, winter, spring, and summer, lysimeter 1 evaporates more on average +18%, +37%, +18%, and +87%, respectively. For the three years, it is estimated that more than +31% of ET occurs from a system with IWS compared to those without IWS (3 and 4). These differences are more noticeable in summer, when the water stored in the IWS allows higher soil moisture during dry and hot periods to be maintained.

In Figure 7, the ET in lysimeters 3 or 4 is lower than the references (1 and 6) during a summer period (24 June to 3 July). A same dynamic and quantity can be observed between the water lost from the storage (dH) and the ET in standard lysimeters. In terms of cumulus of ET and water changes (dH) for these 10 days are ET1 = 47 $\pm$ 2 mm, ET6 = 68 $\pm$ 2 mm, ET3 = 27 $\pm$ 1 mm, ET4 = 31.4 $\pm$ 1mm, dH1 = $-41\pm$ 2 mm, and dH6 = $-58\pm$ 2 mm. In this dry period without rain and exfiltration, for standard lysimeters 1 and 6, the water in the IWS contributes to evapotranspiration by 87 $\pm$ 7% and 85 $\pm$ 5%, respectively. However, in lysimeter 3, ET does not occur at the potential rate and is therefore limited by the water availability.

- The effect of vegetation

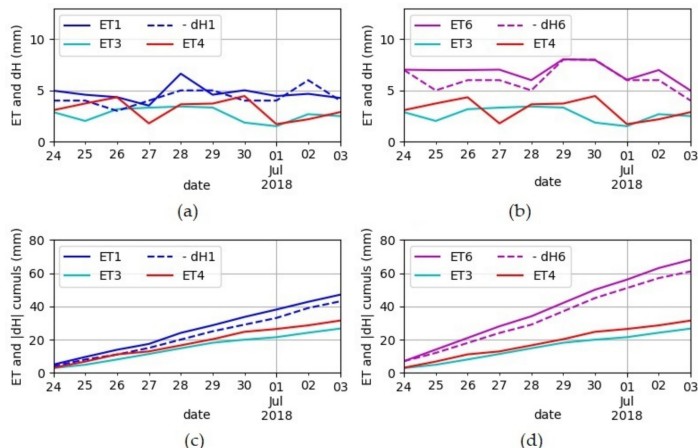

(a)    (b)

(c)    (d)

**Figure 7.** Evapotranspirations (ETs) and water level variations (dH) in the internal water storage for lysimeters 1, 3, and 4 (**a**) and lysimeters 6, 3, and 4 (**b**) during a summer period (24 June to 3 July 2018). (**c,d**) show cumulative values for ETs and $|dH|$, respectively, for lysimeters 1, 3, 4, and lysimeters 6, 3, and 4.

Four types of vegetation (herbaceous, shrubs, spontaneous, and removed vegetation) were tested. A comparison of herbaceous plants (1, 6) with shrubs (2, 7) was problematic because while the former can be considered as acceptable replicas, the latter cannot. A priori, it might be possible to compare them according to their closeness. For example, lysimeters 1 vs. 2, and lysimeters 6 vs. 7 are couples that can be used to identify potential differences between herbs and shrubs. Statistically with the Wilcoxon test, on the whole data, there is no difference between lysimeter 1 and 2 distributions except for the fall season (Table A2). For lysimeter 6 and 7, their distributions differ statically for the whole data set. The role of the herbaceous/shrubby vegetation type seems to be difficult to show based on the whole data for the three years.

In terms of cumulative amounts during all period, the spontaneous vegetation (lysimeter 5) produced a lower ET of 4% than the references (difference not significant according to the Wilcoxon test except for fall and winter). However, if we compare quantitatively by season (Tables 5 and A6), the spontaneous vegetation evaporates more than the references in Summer and Spring even if its maximum values are lower than those of lysimeter 1. That is why in the previous comparison in Table 5 (where spring and summer data were dominant) spontaneous vegetation was more important in terms of cumulated ET. Another term to be taken into account in this comparison is the evolution of vegetation. In the first year, for all seasons, lysimeter 1 (reference) shows a higher evaporation while, in the other two drier years (2017–2018 and 2018–2019), the lysimeter 5 (spontaneous vegetation) evaporates more in spring and summer. This could be explained by the fact that spontaneous vegetation adapts more in these periods of water limitations compared to other vegetation. Moreover, the spontaneous vegetation was not well established at the beginning of the experiment and it developed strongly later (Figures A1 and A2). Table A6, which compares the common days between the three years, shows this point. For the summer period 2018 (23 June–4 July), a higher evapotranspiration of spontaneous vegetation is observed confirming the above results (Figure 8a).

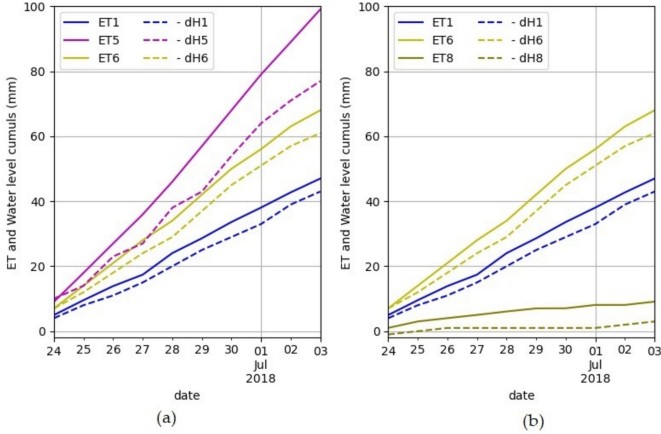

**Figure 8.** Cumulative curves during a summer period (24 June to 3 July 2018) for evapotranspirations (ET) and water level variations (dH) in the internal water storage (IWS). The herbaceous lysime-ters (1 and 6) are compared with the spontaneous vegetation lysimeter (5) (**a**) and, the regularly removed vegetation one (lysimeter 8) (**b**). Note that for this period, the data for shrubs (2, 7) are not valid.

Finally, the regularly removed vegetation (lysimeter 8) produced a lower evapotranspiration of about −17% than lysimeter 1. Statistically, this difference exists globally and would be more pronounced in autumn, spring, and summer. In the summer of 2018, as the vegetation was removed on 21 June, the difference (ET, dH) is more significant (Figures 8b, A1 and A2) and showing that the plants need to develop sufficiently to be able to properly use the water stored in the IWS.

- Meteorological factors

Apart from the factors related to gardens properties, ET is also subject to meteorological factors. In Table 6, the Pearson coefficients (ratio of covariance to the product of the standard deviations) give an overview of the linear correlation between each estimated daily ET and the measured atmospheric variables.

**Table 6.** Linear correlation coefficients (Pearson) between estimated evapotranspiration (ET) and measured meteorological variables.

| Lysimeters | ET1 | ET2 | ET3 | ET4 | ET5 | ET6 | ET7 | ET8 |
|---|---|---|---|---|---|---|---|---|
| $R_G$ (MJ/d) | 0.44 | 0.30 | 0.42 | 0.59 | 0.68 | 0.59 | 0.37 | 0.42 |
| $T$ (°C) | 0.38 | 0.39 | 0.29 | 0.41 | 0.50 | 0.48 | 0.42 | 0.38 |
| $H_R$ (%) | −0.25 | −0.12 | −0.21 | −0.36 | −0.46 | −0.36 | −0.21 | −0.3 |
| $u_w$ (m/s) | 0.05 | 0.08 | −0.08 | −0.2 | −0.06 | 0.05 | 0.09 | 0.02 |
| $P_{atm}$ (hPa) | −0.17 | −0.23 | −0.08 | −0.03 | −0.08 | −0.13 | −0.19 | −0.16 |

A positive correlation between ET and the variables of global solar radiation (0.30 to 0.68) and mean air temperature (0.29 to 0.48) is observed. However, this correlation is of the same order but negative for air humidity (−0.46 to −0.25), and weak for wind speed (−0.06 to 0.08) and atmospheric pressure (−0.23 to 0.08).

A more detailed analysis with PSL models confirms that for all measuring devices (lysimeters and evaporimeter), air temperature and global radiation are the most important variables influencing evapotranspiration with a VIP score greater than one (Figure 9). Moreover, air humidity has a moderate influence on ET in rain gardens (VIP between 0.8 and 1), but for the evaporimeter, it appears as an important determinant for the process (VIP > 1). As previously, wind speed and atmospheric pressure seem to be of low importance. In synthesis, the main atmospheric factors that impact the ET process in these devices are global radiation, air temperature, and air humidity.

- The impact of shading

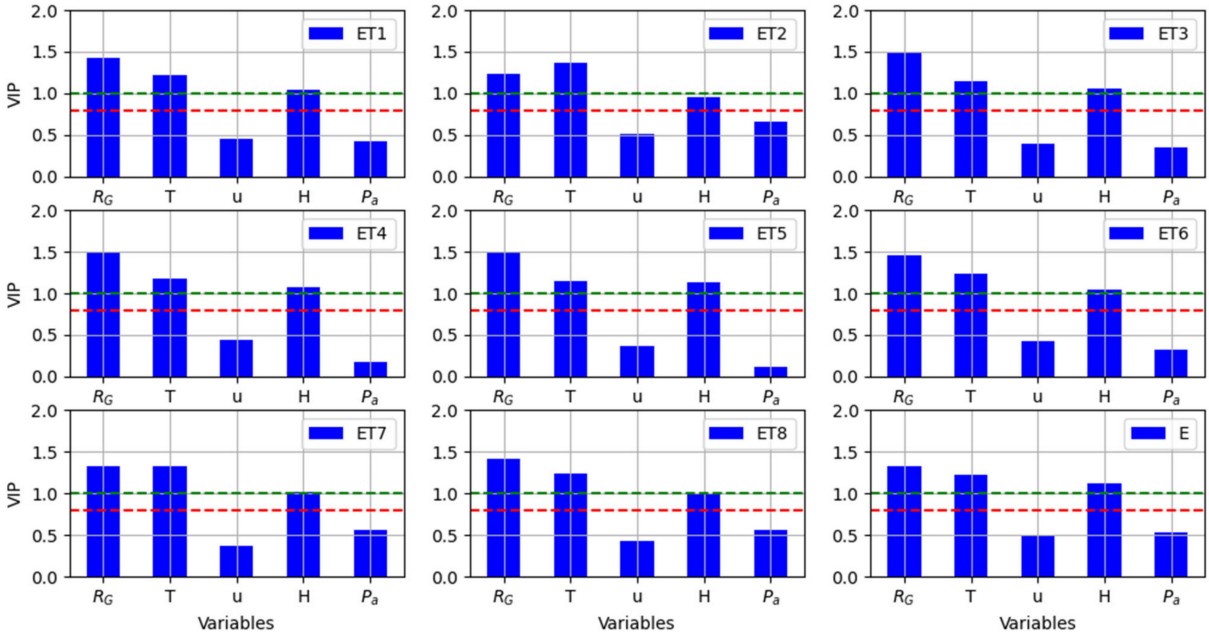

**Figure 9.** Variable importance in projection (VIP) plots according to partial least squares analysis for lysimeters and evaporimeter. The red and green dashed lines refer respectively to the values of VIP larger than 0.8 and 1. Analysis is conducted based on common data between the explained variables and the evaporation (E) or the lysimeter evapotranspiration (ET).

Another factor that needs to be addressed is the exposure of each lysimeter. The «Rain», which is used to estimate the ET, is susceptible to be impacted by the lysimeter exposition. The closeness and height of the south wall could act as a barrier to rain during a windy period. In such conditions, certain lysimeters could receive less rain than others and so does the rain gauge. Thus, the rainfall measurements from lysimeter 7 and 8 are less consistent with the rain gauge measurements (Figure A3). As a result, the further away from the rain gauge the less important the determination coefficient of the regression model is (Figure A3). Some of other atmospheric conditions can be different between lysimeters, in particular due to the buildings and more precisely the wall in the south, which is close to lysimeters 7 and 8 (less than 1 m). Two main variables can be mentioned:

- The global incident radiation is modified by the evolution of the shading. The shading is variable on the lysimeters, both during the day and seasonally. Shading effect is clearly visible in summer around mid-afternoon on the global radiation measurement with strong decrease in the value (Figure A4);
- The Wind: the linear correlation between wind speed and ET is found to be weak (Table 6). However, the wind could impact the distribution of rainfall on the lysimeters.

These potential modifications would reduce the ET on the southern lysimeters (7 and 8), compared to those to the north. This impact is difficult to assess quantitatively as it would require specific measurements (e.g., 3D site geometry) or extended replicas. Regarding the shrub configuration, it could be suggested that the exposition effect is responsible for the great differences observed between the replicas (lysimeters 2 and 7) that are opposite to each other. Therefore, this difference is estimated on the cumulative ETs (+11%) and also on the scatterplot of the daily ETs (Figure 6c).

*3.3. Evapotranspiration Predictive Equations, ETs Estimated from the Evaporimeter, and the Lysimeters*

The objective of this section is to compare ET estimated by water budget on lysimeters to two types of reference values: (i) evaporation measured on an open water surface with the pan evaporimeter (Figure 4i) and (ii) ET estimated with potential formulations. In order to increase the amount of available data, the numbers of lysimeters were reduced to the different configurations (1, 2, 3, 5, and 8) and the common validated data over the whole period of study are for 281 days (Table 7).

**Table 7.** Comparative totals and averages (in mm) of evaporation (E) and evapotranspiration (ET) estimated, respectively, with the evaporimeter and the lysimeters (data = 281).

| Seasons (Data) | ET1 | ET2 | ET3 | ET5 | ET8 | E |
|---|---|---|---|---|---|---|
| Fall (53) | $93 \pm 4$ | $137 \pm 3$ | $62 \pm 3$ | $100.0 \pm 4$ | $64 \pm 3$ | 40.1 |
| Winter (14) | $34.3 \pm 2$ | $28 \pm 2$ | $24 \pm 2$ | $14 \pm 2$ | $16 \pm 2$ | 13 |
| Spring (81) | $311 \pm 5$ | $294 \pm 4$ | $290 \pm 4$ | $330 \pm 5$ | $27 \pm 4$ | 180 |
| Summer (133) | $551 \pm 6$ | $570 \pm 5$ | $370 \pm 5$ | $637 \pm 6.3$ | $476 \pm 5$ | 382.4 |
| Cumulus (281) | $988 \pm 9$ | $1029 \pm 8$ | $746 \pm 8$ | $1081 \pm 9$ | $836 \pm 8$ | 585 |
| Mean | 3.5 | 3.6 | 2.6 | 3.8 | 2.9 | 2.1 |

Cumulative ETs indicate that evaporation from the water surface is 585 mm (average 2.1 mm/d), while ET from the lysimeters varies from 746 to 988 mm (average 2.6 and 3.5 mm/d). Compared to the lysimeter 1, this represents a difference of −41%. At the daily step, the E of the evaporimeter is almost systematically lower than the ET of the lysimeters; this is also the case for high values (>8 mm). Compared to the non-IWS configuration (lysimeter 3), the trend is much less marked mainly in summer (when the evaporimeter evaporates more, 382 vs. 370 mm).

Here, the results indicate that the evaporation of the free water surface near the lysimeters is low compared to the ET for rain gardens. This result is not intuitive because the open water surface is always supplied with water and does not offer theoretically a resistance to the ET flux. One hypothesis is that the development of the plants in the lysimeters leads to a larger evapotranspiration surface than what is theoretically perceived, i.e., 1 m². Indeed, the surface for evaporation and transpiration for lysimeters is larger than the evaporimeter, so that under certain conditions with no hydric limitations, ET in lysimeters is more important. However, in summer, where the lysimeter 3 does not have IWS, the quantity of evapotranspiration is reduced compared to the evaporimeter. If a short dry period is considered (as mentioned above, 24 June–3 July 2018), the evaporated water from the evaporimeter is slightly higher than the lysimeter 1, which has an IWS (in terms of accumulation 50 ± 2 and 47 mm, respectively, for evaporimeter and lysimeter 1 Figure 10c).

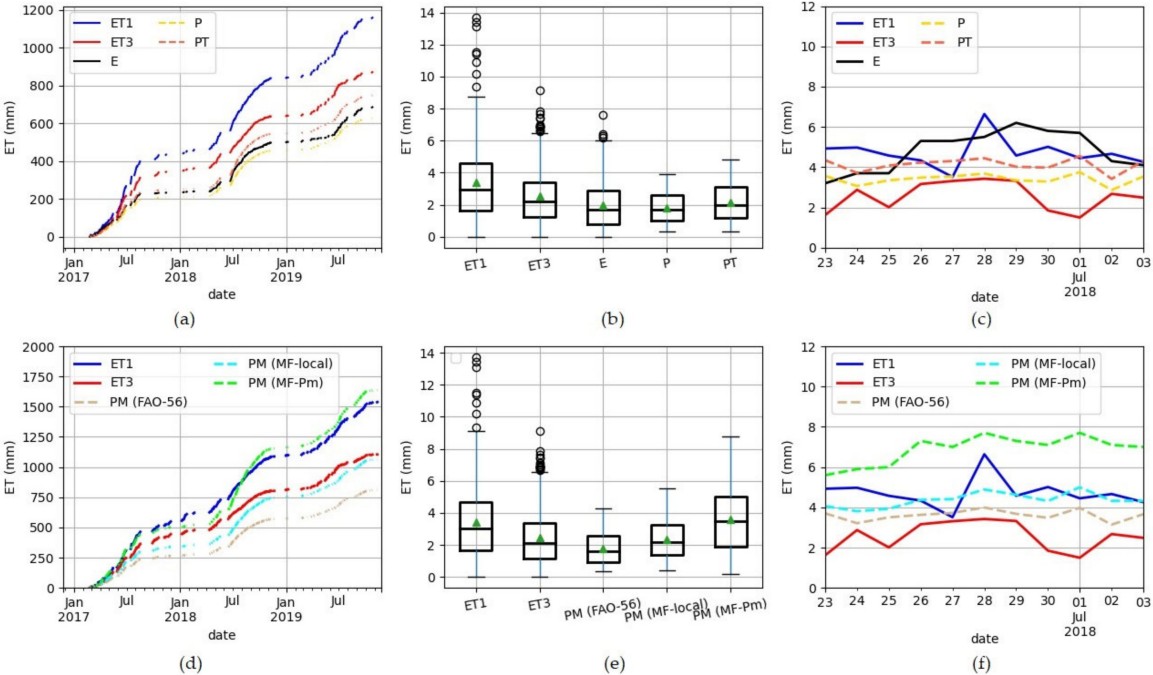

**Figure 10.** Evapotranspirations (ET) from the lysimeter 1(with internal water storage), the lysimeter 3 (without internal water storage), the evaporimeter, and the potential ET models: (**a–c**) compare the Penman (P) and Priestley–Taylor (PT) potential ETs to ET1 and ET3 (data = 346) and respectively present the cumulative ET values, the boxplots, and the ET dynamics for a dry period (24 June to 3 July 2018); (**d–f**) refer to the ET1, ET3, Potential ET of FAO-56, and Meteo-France models (data = 453) and show, respectively, cumulative ET values, boxplots, and ET dynamics for a dry period (24 June to 3 July 2018). PM and Pm refer respectively to Penman–Monteith and Paris-Montsouris.

The reference (lysimeter 1, with IWS), as well as the lysimeter 3 without IWS are used in comparison with the models because of their closeness to the weather station and the interest in observing the validity of models regarding to the storage presence or not. PT, P, PM (Fao-56), and PM (MF-local.) are potential ETs evaluated with the local meteorological data while PM (MF-Pm) is estimated from the Météo-France equation at the Paris-Montsouris (Pm) station, approximately 2.5 km from the site (Figure 10; Table 8). At Paris-Montsouris, the station is clear and far from the obstacles that can affect the meteorological measurements.

Over the whole study period, ET on the Paris-Montsouris station PM (MF-Pm) is more important in terms of trend and cumulative amount (1639 mm) than the ET estimated locally (1066 mm) with the same Météo-France formulation (Table 8 and Figure 10c–e). This observation illustrates the impacts of the microclimatic variabilities on the assessment of potential ET in urban area. These variations are mainly due to the global incident radiation

variable, which over the three years is on average 135 and 80 w/m$^2$, respectively, at Paris-Montsouris and at the study site. Compared to lysimeters (1, 3) and other models, PM (MF-Pm) is more important particularly in summer and spring.

**Table 8.** Cumulatives and averages (in mm) evapotranspirations (ET) obtained with the lysimeters (ET1 and ET3) and the potential ET models. P, PT, PM (Fao-56), PM (MF-local.) are Penman, Priestley–Taylor, Fao-56, and Météo-France potential ETs evaluated with the local meteorological data while PM (MF-Pm) is the potential ET evaluated with the Paris-Montsouris station data.

| Seasons (Data) | ET1 | ET3 | P | PT | PM (FAO-56) | PM (MF-local) | PM (MF-Pm) |
|---|---|---|---|---|---|---|---|
| Fall (109) | 240 ± 6 | 154 ± 5 | 78 | 89 | 93 | 141 | 199 |
| Winter (55) | 130 ± 4 | 65 ± 3 | 32 | 36 | 39 | 58 | 75 |
| Spring (134) | 543 ± 6 | 475 ± 5 | 260 | 312 | 283 | 353 | 536 |
| Summer (155) | 646 ± 7 | 414 ± 6 | 364 | 437 | 396 | 515 | 829 |
| Cumulus (453) | 1559 ± 11 | 1107 ± 10 | 734 | 874 | 811 | 1066 | 1639 |
| mean | 3.4 | 2.4 | 1.6 | 2 | 1.8 | 2.4 | 3.6 |

For local potential evapotranspirations, i.e., PM (FAO-56), PM (MF-local), P, and PT, a general underestimation (systematically in fall and winter) of lysimeters (1, 3) ETs is observed over the whole simulation period (Figure 10). However, PM (MF-local) seems to be a reasonably good predictor of the ET when IWS is absent (Table 8).

For the June 24 to 3 July 2018 dry period, potential ETs are superior to lysimeter 3 and lower than lysimeter 1, which evaporates and transpires at a considerable rate (Figure 10c,f).

The evaporimeter data (346 observations) were also be compared with potential ET values from Penman and Priestley–Taylor equations. In general, the Priestley–Taylor model (749 mm) overestimates the evaporimeter measurement (686 mm), and the Penman model (627 mm) underestimates it. While the maximum values are significant for the evaporimeter, in terms of average, the three estimates are close (1.9 mm, 1.8 mm, and 2.1 mm, respectively, for E, P and PT). The sensitivity of the Priestley–Taylor equation to the value $\alpha_{PT}$ [39] suggests that modelling ET or E from this equation requires a sensitivity study that would lead to a specific $\alpha_{PT}$ value (and that is not the objective here).

## 4. Discussion

In this study, the estimated ETs in the pilot rain gardens account for 61–90% of the collected rainfall (Table A3). They are on average 2.4 to 3.78 mm/d, depending on the vegetation maintenance, the presence (3–3.78 mm/d) of an IWS or not (2.4–2.5 mm/d). The uncertainties of ETs are ±0.42 to ±0.58 mm. A similar experiment conducted at the Villanova University in Pennsylvania (USA) by Wadzuk et al. [13] showed ET means of 6.1 and 3.1 mm/d, respectively, for IWS or no IWS lysimeters from April to November in 2010 and 2011. The high value of the lysimeter with an IWS compared to what is found here could be explained by the focalised summer period, by the size of the IWS, which is larger in the Wadzuk case (36 cm), by the evaporative demand, and also the inputs (precipitations) to the lysimeter. However, our values are in the same range as those estimated by Hess et al. [12] (4.3 and 2.7–2.9 mm/d, respectively, for IWS or no IWS lysimeters). These two studies showed the importance of ET in the rain gardens and estimate it between 43 and 78% of the collected rainfall.

The impact of the vegetation type (herbaceous and shrubs) was not addressed here because the replicas of shrubs (lysimeters 2 and 7) showed distributions that were statistically significant and there were large differences in terms of cumulative amounts. It has been suggested that this situation is probably due to the shading, which has a great impact on lysimeter 7 by significantly reducing its ET. However, a comparison of the closely related lysimeters 1 and 2 shows a higher evapotranspiration of shrubs configuration (+6%

globally and more significant in autumn based on the Wilcoxon test), which suggests that the vegetation type may be a factor to be considered. Indeed, Nocco et al. [14] investigated the impact of vegetation and vegetation type (grassland, shrubs, turf, and bare soil) in the hydrological performance of free-draining rain gardens in the Midwest of the USA. Their studies were conducted in three months (July, August, and September) and they first found that the effect of vegetation is significant when evaporative demand is also high. In addition, the configuration with grassland showed a higher ET than the others with an average of 9 and 7 mm/d in August and July, respectively. The ET of the bare soil system was lower (4 and 3 mm/d in August and July, respectively), and the ET of the shrubs, in contrast to the other systems that had ETs that decreased as the evaporative demand decreased, had a relatively constant ET around 6 mm/d for the months of July and August. It can be seen that the type of plant not only significantly impacts the ET dynamics, but also the ET cumulative values.

Another factor that was not tested in this study is the effect of soil type. Hess et al. [12] estimated average ETs of 2.9 and 2.7 mm/d, respectively, for free-draining lysimeters with local sandy loam and sandy soil (three-year data). In terms of water balance, the ET with the local soil accounted for 47% and the other 43% of the water balance. These differences show that finer soils are more favourable to ET as they retain much more water than coarse textured soils [15]. However, the issue of soil impact is best addressed in conjunction with the type of vegetation, as vegetation through its development may affect the hydraulic properties of the soil, which are able to affect the ET. Johnston's [40,41] and Le Coustumer et al. [42] illustrate the link between soil and vegetation evolution [14]. The first showed that grassland and shrub rain gardens (without IWS) have significantly lower volumetric soil water content at depths of 0–0.15 and 0.30–0.45 m (3–4 and 10% lower, respectively) compared to turf rain gardens prior to storms, suggesting that vegetation type can impact on the storage capacity of rain gardens. The second one indicated that the type of vegetation through the growth and the morphology of their roots impact the hydraulic conductivity of the soil (with average hydraulic conductivity decreasing by a factor of 3.6 over the 72 weeks of testing) that influences mainly the drainage and the water availability in the garden. For example, Le Coustumer et al. [42] observed that a species with thick roots significantly maintained the permeability of the soil over time. This issue of the link between plants, and soil in these systems, is not limited to the sustainable hydrological services but may well extend to the sustainability of other ecological services (e.g., removal of pollutants, see Glaister et al. [43]) in SUDS.

Until now, it can be argued that if the aim of rain garden design is to maximise ET, it needs to provide an underlying water storage, and select a balanced choice between the vegetation type and the soil type. Further factors to consider are atmospheric factors as shown in the PSL approach, global radiation, air temperature, and humidity impact on the estimated ET of lysimeters. Such factors are responsible for the seasonal variations in ET flux. Similar to the other GIS (e.g., green roofs with Feng et al. [36]), these three variables are known to affect ET and are generally input variables for the models used to simulate ET. Understanding the impacts of atmospheric variables on rain garden ET requires suitable hydrological simulation tools.

Finally, for a better efficiency of hydrological models, the ET process should be well represented. In fact, the ET prediction equations used in these models are based on the concept of potential ET, which account for evaporative demand. These ET models are then coupled with specificities related to vegetation, water availability, and/or local microclimatic conditions (FAO methods [23], WUCOLS [44], and LIMP [45] methods). However, these methods have remained impractical [15,17,27,46–48] as they require measurements of multiple parameters, are derived from the agricultural context, and are less suitable for the urban context. Hess et al. [27] tested the validity of the ASCE-Penman–Monteith [24] and Hargreaves equations in rain garden systems (one system with storage and two systems without storage with free drainage). Without including crop coefficients (estimated ET divided by potential ET) and soil moisture extraction functions, these equations provided

an adequate estimate of rain garden ET for all systems on a storm scale. The use of crop coefficients and soil moisture extraction functions in both equations reduced the errors in the ET estimates and increased the predictive power of the equations for all types of weighing lysimeters at the daily scale. In this study, it is found that with potential ETs (FAO and Météo-France) evaluated with local atmospheric variables, lysimeters (one with IWS and three without IWS) were underestimated by the models. Furthermore, if the input data (atmospheric variables) are not local, as shown in Figure 10, then there are issues of urban micrometeorological variability [47] to take into account. However, at a seasonal scale, the PM (MF-local) equation seems to be a good approximation of the ET of the lysimeter without reserve. Monitoring properties that describe the dynamics of the vegetation canopy (stomatal resistance, LAI, roots expansion, etc.) and soil water content would enable more accurate assessment of the impact on plants and the comparison of lysimeters to evaporate and transpire [19].

## 5. Conclusions

The process of evapotranspiration should be included in the design of green infrastructure systems (GIS) in order to optimise their hydrological functions of stormwater management and their ability to cool the urban area in hot periods. In this study, a comparison of the evapotranspiration capacity between different pilot rain garden configurations, with an impluvium equal to four times the vegetated surface, was carried out, based on data covering a three-year period in Paris (temperate climate, France) that has undergone rigorous validation. The validated periods are less rainy and represent more the summer and spring seasons. It was found that the evapotranspiration flux from rain gardens is significant, with values that can exceed 8 to 12 mm/d in summer period for several days, and is characterized by a marked seasonality with very low values in winter ($\leq 2$ mm/d). The installation of an internal water storage at the base is the most favourable determinant to enhance the flux and reduce exfiltration (+28 to 30% if the reference lysimeter with an IWS and those without IWS are compared). The vegetation, here, is a secondary determinant, and less marked (+6% for shrubs compared the reference herbaceous). The spontaneous flora gives more ET than the reference configuration in summer (+8%) and all configurations evaporate and transpire more than the regularly removed vegetation configuration. The positioning of the lysimeters between them (close to or far from buildings) also seems to be a determining factor and, in particular, the shading, which has a reducing effect on ET (the replica that is less exposed to the shade evaporates 15% more than the shaded one).

The experimental set-up used in this work was pertinent, and allowed the observation of water balance components and the assessment of the multi-annual daily ET with admissible uncertainties ($\pm 0.42$ to 0.58 mm). Therefore, the seasonal dynamics and the relative significance of each determinant of ET in the rain gardens were highlighted. A possible counterintuitive result in the seasonal analysis was also that the ET values observed on the rain gardens, and particularly for those with an IWS, are higher than the ET from an evaporimeter. Based on the potential ET from a reference station located at 2.4 km from the site, the ET is under-estimated for the setup with an IWS during the winter and fall seasons.

Future studies need to include some aspects in the experimental setup for still better understanding the ET process in rain gardens. First, the location of the experiment should be selected in such a way that local microclimatic factors and especially shading effects are taken into account. Second, monitoring some properties, which describe the dynamics of the vegetation canopy (stomatal resistance, LAI, roots expansion, etc.) [19], and a lysimeter without vegetation could be added to experimentally compare the contribution of plant transpiration and soil evaporation. Finally, these results of ET could be used to investigate the modelling of hydrological processes and more especially on the ET process in urban rain gardens. The use of detailed and physically based hydro-climatic models (as SisPAT [49] et Teb-hydro [50]) should make it possible to better understand and reproduce the process. Nevertheless, the use of this type of models requires a large data set for the parametrization and evaluation steps.

**Author Contributions:** A.A.O. realized the analysis of the data and wrote the paper. E.B. supervised the study and revised the paper with M.-C.G., B.D. All authors have read and agreed to the published version of the manuscript.

**Funding:** This work was partially funded by the Paris City Council (grant n° 2019 DPE 57/villede-ParisEJ45026788290), which contributed to the experimental set-up and data acquisition. It was also supported by the OPUR program and the Seine-Normandy Water Agency.

**Institutional Review Board Statement:** Not applicable.

**Informed Consent Statement:** Not applicable.

**Acknowledgments:** Authors are grateful for the support given by M. Ramier David for the data processing, by Md. Tang Jieyu and Bonnefous Hortense for their important contributions to the data analysis and the partners from the Paris City Council(DPE-STEA) and set up the experiment and contributed for data collecting and the OPUR Program.

**Conflicts of Interest:** The authors declare no conflict of interest.

## Appendix A. Assessment of VIP Score from a Partial Least Square (PLS) Model

By resuming the influence of individual X variables on the PLS model, the VIP scores are assessed as the weighted sum of squares of the PLS weights, $w$, which take into account the amount of explained y variance in each extracted latent variable [33,34]. The VIP score for a given variable $j^{th}$ is given according to Farrès et al. [33]:

$$VIP_j = \sqrt{\frac{\sum_{f-1}^{F} w_{jf}^2 SSY_f J}{SSY_{total}.\ F}} \qquad (A1)$$

where $w_{jf}$ is the weight value for $j$ variable and $f$ component, $SSY_f$ is the sum of squares of explained variance for the $f_{th}$ component and $J$ number of X variables, $SSY_{total}$ is the total sum of squares explained of the dependent variable, and $F$ is the total number of components. The $w_{jf}^2$ gives the importance of the $j^{th}$ variable in each $f_{th}$ component, and $VIP_j$ is a measure of the global contribution of $j$ variable in the complete PLS model.

$$SSY_f = b_t^2 t'_f t_f SSY_{total} = b^2 T'T. \qquad (A2)$$

where $T$ is the $X$ scores matrix and $b$ is the PLS inner relation vector of coefficients.

## Appendix B. Figures

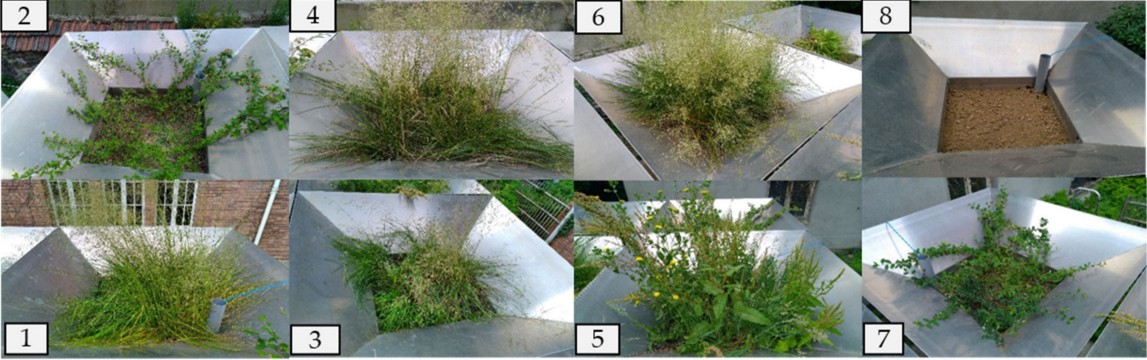

**Figure A1.** The vegetation in the eight lysimeters on 21 June 2018 (Source: DPE-STEA, Paris council). It can be observed that the spontaneous vegetation (lysimeter 5) is more developed compared to the other settings and that today the vegetation in lysimeter 8 has been removed. Moreover, the shrub configurations (2,7) are not well developed compared to the other configurations.

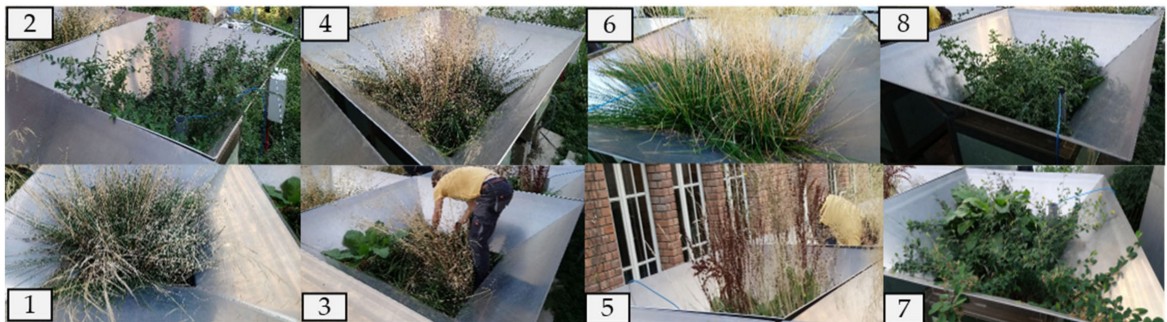

**Figure A2.** The vegetation in the eight lysimeters on 20 September 2018 (Source: DPE-STEA, Paris council). It can be observed particularly a development of the shrubs (lysimeter 2 and 7) and a revegetation of lysimeter 8 three months after all the plants have been removed.

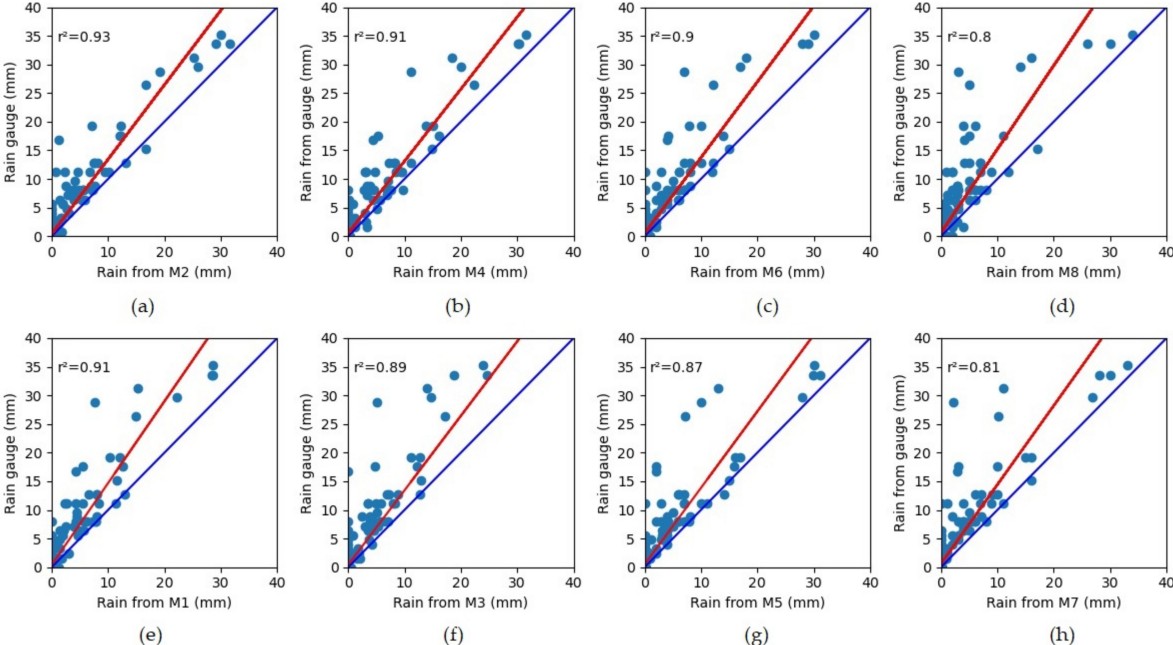

**Figure A3.** Comparison of the rain measurements resulting from the rain gauge data and by the mass variation in the system (rainfall are estimated as any increase in the total mass (lysimeter mass + exfiltration) of the lysimeter). The rainfall estimated from the lysimeter masses is lower compared to the rain gauge. Figure (**a**–**h**) represent, respectively, lysimeters 2, 4, 6, 8, 1, 3, 5, and 7.

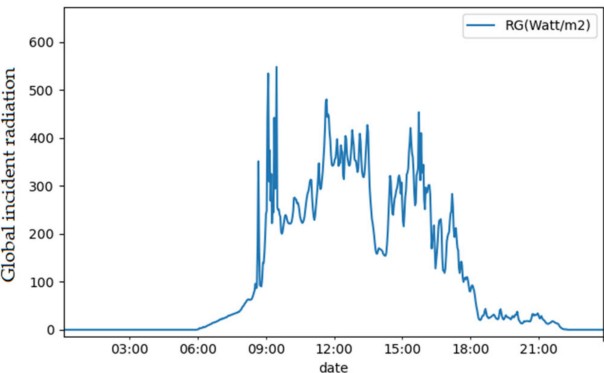

**Figure A4.** Global incident radiation (w/m$^2$) of 14 June 2018 measured at a time step of 2 min.

## Appendix C. Tables

**Table A1.** Validated data after processing for the three years (2016–2019). The numbers and the symbol "E" refer respectively to lysimeters and evaporimeter. For season, it was considered fall (22 September to 21 December), winter (21 December to 20 March), spring (20 March to 21 June), and summer (21 June to 22 September). The total data are for 1096 days.

| Parameter Seasons | | 1 | 2 | 3 | 4 | 5 | 6 | 7 | 8 | E | Rain |
|---|---|---|---|---|---|---|---|---|---|---|---|
| ET | Fall | 151 | 197 | 181 | 186 | 166 | 161 | 163 | 207 | 126 | 239 |
| | Spring | 145 | 145 | 169 | 169 | 151 | 131 | 154 | 173 | 143 | 198 |
| | Summer | 161 | 198 | 203 | 198 | 192 | 192 | 199 | 214 | 185 | 246 |
| | Winter | 124 | 152 | 101 | 111 | 138 | 146 | 150 | 151 | 74 | 224 |
| | Total | 581 (53%) | 692 (63%) | 654 (59%) | 664 (60.6%) | 647 (59%) | 630 (57.5%) | 666 (60.8%) | 745 (68%) | 528 (48.2%) | 907 (82.7%) |
| Exf | Fall | 197 | 231 | 215 | 215 | 183 | 200 | 218 | 231 | - | |
| | Spring | 213 | 207 | 229 | 221 | 200 | 161 | 192 | 217 | - | |
| | Summer | 195 | 232 | 246 | 249 | 230 | 235 | 251 | 251 | - | |
| | Winter | 174 | 196 | 149 | 161 | 163 | 202 | 196 | 209 | - | |
| | Total | 779 (71%) | 866 (79%) | 839 (76.6%) | 846 (77.2%) | 776 (71%) | 798 (73%) | 857 (78.2%) | 908 (83%) | | - |
| $ds$ | Fall | 196 | 208 | 205 | 192 | 213 | 202 | 182 | 217 | - | |
| | Spring | 186 | 195 | 193 | 193 | 195 | 197 | 200 | 197 | - | |
| | Summer | 214 | 215 | 213 | 203 | 217 | 211 | 204 | 221 | - | |
| | Winter | 190 | 202 | 190 | 143 | 209 | 180 | 189 | 181 | - | |
| | Total | 786 (71.8%) | 820 (75%) | 801 (73%) | 731 (66.7%) | 834 (76%) | 790 (72%) | 775 (70%) | 816 (74%) | | - |
| $dL$ | Fall | 197 | 193 | - | - | 215 | 197 | 196 | 212 | - | |
| | Spring | 198 | 188 | - | - | 164 | 191 | 188 | 196 | - | |
| | Summer | 200 | 204 | - | - | 204 | 174 | 214 | 205 | - | |
| | Winter | 175 | 182 | - | - | 213 | 209 | 189 | 202 | - | |
| | Total | 770 (70.2%) | 767 (70%) | - | - | 796 (72.6%) | 771 (70.3%) | 787 (71.8%) | 815 (74.4%) | | - |

**Table A2.** A Wilcoxon test (wt) results comparing lysimeters replicas and different configurations to the reference one the lysimeter 1. The test is performed based on the whole validated data and seasons. For the seasons, it is considered fall (22 September to 21 December), winter (21 December to 20 March), spring (20 March to 21 June), and summer (21 June to 22 September). Note that the coloured boxes represent *p*-values, which are superior to 5%.

| | Lysimeters | | Validated Data | Seasonal Comparison | | | |
|---|---|---|---|---|---|---|---|
| | | | | Fall (273 Days) | Winter (272 Days) | Spring (276 Days) | Summer (276 Days) |
| | | | $p_v$ | $p_v$ | $p_v$ | $p_v$ | $p_v$ |
| Comparison of the Replicas | 1, 6 | wt | 0.87 | 0.35 | 0.333 | 0.42 | 0.511 |
| | | n | (445) | (112/273) | (82/272) | (106/276) | (145/276) |
| | 3, 4 | wt | 0.007 | 0.007 | 0.004 | 0.05 | 0.26 |
| | | n | (600) | (160/273) | (87/272) | (166/276) | (187/276) |
| | 2, 7 | wt | $2 \times 10^{-16}$ | $8.45 \times 10^{-7}$ | $1.72 \times 10^{-6}$ | 0.08 | $1.2 \times 10^{-06}$ |
| | | n | (583) | (154/273) | (116/272) | (127/276) | (186/276) |
| Different settings compared to the reference (1 or 6) | 1, 3 | wt | $1.16 \times 10^{-20}$ | $4.5 \times 10^{-7}$ | 0.0001 | 0.01 | $5 \times 10^{-12}$ |
| | | n | (464) | (113/273) | (58/272) | (135/276) | (186/276) |
| | 1, 4 | wt | $1.15 \times 10^{-62}$ | $4.14 \times 10^{-20}$ | $1.04 \times 10^{-15}$ | $3.98 \times 10^{-12}$ | $1.01 \times 10^{-19}$ |
| | | n | (460) | (117/273) | (57/272) | (133/272) | (153/272) |
| | 1, 5 (data) | wt | 0.1 | 0.0061 | $1.78 \times 10^{-6}$ | 0.15 | 0.17 |
| | | n | (475) | (121/273) | (86/272) | (118/276) | (150/276) |
| | 1, 2 | wt | 0.098 | 0.046 | 0.3844 | 0.81 | 0.36 |
| | | n | (503) | (136/273) | (89/272) | (121/276) | (157/276) |
| | 1, 8 | wt | $2 \times 10^{-8}$ | $8 \times 10^{-5}$ | 0.134 | $2 \times 10^{-4}$ | 0.01 |
| | | n | (515) | (136/273) | (86/272) | (134/276) | (159/276) |
| | 6, 7 | wt | 0.001 | 0.59 | 0.288 | $5 \times 10^{-6}$ | 0.97 |
| | | n | (538) | (125/273) | (111/272) | (123/276) | (179/276) |
| | 6, 5 | wt | 0.08 | 0.06 | 0.0037 | 0.89 | 0.34 |
| | | n | (506) | (122/273) | (94/272) | (122/276) | (168/276) |

**Table A3.** Proportions of evapotranspiration (ET) to the rainfall (4P) received in each lysimeter. The number of data (n) refers to the common validated data between the rainfall and the considered ET of the lysimeter (on a total of 1096 days).

| Lysimeter (n) | 1 (557) | 2 (674) | 3 (636) | 4 (630) | 5 (624) | 6 (601) | 7 (656) | 8 (718) |
|---|---|---|---|---|---|---|---|---|
| ET (mm) | 1705.7 | 2295.9 | 1605.3 | 1491.5 | 1846.3 | 1846 | 1919.9 | 1924.6 |
| P (mm) | 502 | 637.6 | 643.8 | 613.8 | 650.6 | 614.8 | 556.6 | 752.4 |
| 4P (mm) | 2008 | 2550.4 | 2575.2 | 2455.2 | 2602.4 | 2459.2 | 2226.4 | 3009.6 |
| %ET | 85% | 90% | 62% | 61% | 71% | 75% | 86% | 64% |

**Table A4.** Cumulative ET in mm for the lysimeters in common validated days (305/1098).

| Year | Seasons | Days | ET1 | ET2 | ET3 | ET4 | ET5 | ET6 | ET7 | ET8 |
|---|---|---|---|---|---|---|---|---|---|---|
| 2016 to 2017 | Fall | 12 | 28.9 ± 1.8 | 45.2 ± 1.6 | 21.8 ± 1.6 | 27.01 ± 1.4 | 12.7 ± 1.9 | 21.6 ± 2 | 25.9 ± 1.9 | 20.6 ± 1.5 |
| | Winter | 0 | 0 | 0 | 0 | 0 | 0 | 0 | 0 | 0 |
| | Spring | 27 | 93.1 ± 2.8 | 118.8 ± 2.4 | 82 ± 2.4 | 87 ± 2.2 | 79.7 ± 2.8 | 104 ± 3 | 83.1 ± 2.8 | 91.7 ± 2.3 |
| | Summer | 15 | 76.1 ± 2 | 97.7 ± 1.8 | 60.2 ± 1.8 | 61.8 ± 1.6 | 47.6 ± 2.1 | 70.5 ± 2.2 | 79.5 ± 2.1 | 58.6 ± 1.7 |
| 2017 to 2018 | Fall | 39 | 82.8 ± 3.4 | 80.9 ± 2.9 | 41.1 ± 2.9 | 36.8 ± 2.6 | 89.2 ± 3.4 | 70 ± 3.6 | 55 ± 3.4 | 44.3 ± 2.8 |
| | Winter | 15 | 45.8 ± 2 | 39.7 ± 1.8 | 23.4 ± 1.8 | 15.2 ± 1.6 | 10.1 ± 2.1 | 14.5 ± 2.2 | 22.8 ± 2.1 | 22.8 ± 1.7 |
| | Spring | 25 | 108.2 ± 2.7 | 88.6 ± 2.3 | 88 ± 2.3 | 90.1 ± 2.1 | 137 ± 2.7 | 135.9 ± 2.9 | 83.2 ± 2.7 | 110 ± 2.2 |
| | Summer | 70 | 234.8 ± 4.5 | 229.9 ± 3.9 | 169 ± 3.9 | 160 ± 3.5 | 321.3 ± 4.6 | 300 ± 4.8 | 205.7 ± 4.6 | 184.9 ± 3.7 |
| 2017 to 2018 | Fall | 14 | 44.7 ± 2 | 48.6 ± 1.7 | 22.7 ± 1.7 | 25.9 ± 1.5 | 35 ± 2 | 53.9 ± 2.1 | 45.7 ± 2 | 35.26 ± 1.7 |
| | Winter | 10 | 22.07 | 14.3 ± 1.7 | 10.9 ± 1.5 | 9.03 ± 1.5 | 19.09 ± 1.7 | 24.8 ± 1.8 | 20.63 ± 1.7 | 9.5 ± 1.4 |
| | Spring | 31 | 122.14 ± 3 | 99 ± 2.6 | 103.9 ± 2.6 | 89.44 ± 2.3 | 127.7 ± 3 | 114.5 ± 3.2 | 117.5 ± 3 | 53.5 ± 2.5 |
| | Summer | 47 | 207.3 ± 3.7 | 218.8 ± 3.2 | 116 ± 3.2 | 147.7 ± 2.8 | 271.4 ± 3.7 | 150.3 ± 3.9 | 222.3 ± 3.7 | 232.2 ± 3.1 |

**Table A5.** Cumulative ET and P in mm for the lysimeters in common validated days for reference (herbaceous) and shrubs configurations. The couples' lysimeter 1 vs. lysimeter 2 and lysimeter 1 vs. lysimeter 7 are presented.

| Year | Seasons | Days (1 vs. 2) | Rain (P) | ET1 | ET2 | Days (1 vs. 7) | Rain (P) | ET1 | ET7 |
|---|---|---|---|---|---|---|---|---|---|
| 2016–2017 | Fall | 40 | 25.8 | 71.17 | 103.98 | 42 | 23.4 | 67.61 | 65.57 |
| | Winter | 38 | 28.8 | 37.36 | 74.68 | 49 | 55 | 57.31 | 34.00 |
| | Spring | 57 | 59.8 | 212.24 | 274.06 | 51 | 87 | 209.13 | 189.99 |
| | Summer | 36 | 37.8 | 193.27 | 216.7 | 34 | 39.2 | 155.91 | 158.99 |
| 2017–2018 | Fall | 61 | 58.4 | 110.44 | 125 | 50 | 26.6 | 97.09 | 70.02 |
| | Winter | 33 | 41.6 | 87.6 | 80 | 29 | 39.6 | 82.01 | 38.02 |
| | Spring | 26 | 20.4 | 112.76 | 90 | 31 | 23.6 | 129.87 | 100.30 |
| | Summer | 72 | 17.6 | 240 | 232 | 72 | 17.6 | 239.94 | 208.93 |
| 2018–2019 | Fall | 33 | 54 | 80.15 | 78.89 | 28 | 52.4 | 68.71 | 96.95 |
| | Winter | 18 | 5.4 | 33.53 | 22.21 | 24 | 10.2 | 37.74 | 36.18 |
| | Spring | 38 | 19.8 | 157.63 | 129.17 | 41 | 15.4 | 179.63 | 175.68 |
| | Summer | 49 | 17.2 | 217.12 | 230.57 | 49 | 17.2 | 217.12 | 230.03 |
| 2016–2018 | Fall | 136 | 138.8 | 264.47 | 315.93 | 121 | 103 | 233.96 | 234.83 |
| | Winter | 89 | 75.8 | 158.5 | 176.89 | 102 | 104.8 | 177.06 | 108.21 |
| | Spring | 121 | 100 | 482.64 | 493.24 | 123 | 126 | 518.63 | 465.97 |
| | Summer | 157 | 72.6 | 650.33 | 679.27 | 155 | 74 | 612.97 | 597.95 |
| Total | | 503 | 387.2 | 1555.95 | 1665.34 | 501 | 407.8 | 1542.63 | 1406.97 |

**Table A6.** Cumulative ET and P in mm for the lysimeters in common validated days for reference (lysimeter 1), spontaneous (lysimeter 5), and regularly removed (lysimeter 8) vegetation configurations.

| Year | Seasons | Days (1 vs. 5) | P | ET1 | ET5 | Days (1 vs. 8) | P | ET1 | ET8 |
|------|---------|------|------|------|------|------|------|------|------|
| 2016–2017 | Fall | 45 | 42 | 81.0 | 35.8 | 39 | 25.8 | 69.9 | 51.7 |
| | Winter | 38 | 51.6 | 43.7 | 18.1 | 33 | 23.6 | 28.2 | 43.8 |
| | Spring | 45 | 66.4 | 187.4 | 162.1 | 60 | 66.8 | 227.3 | 224.2 |
| | Summer | 30 | 22.2 | 163.3 | 88.4 | 38 | 39.2 | 205.7 | 156.0 |
| 2017–2018 | Fall | 61 | 60.2 | 112.4 | 113.6 | 61 | 50.8 | 114.5 | 72.5 |
| | Winter | 34 | 44.8 | 89.1 | 22.8 | 30 | 36.8 | 84.8 | 55.0 |
| | Spring | 32 | 23.6 | 134.4 | 165.8 | 32 | 23.6 | 134.4 | 139.3 |
| | Summer | 72 | 17.6 | 239.9 | 323.8 | 72 | 17.6 | 239.9 | 189.4 |
| 2018–2019 | Fall | 15 | 35 | 48.4 | 35.9 | 35 | 56.4 | 79.3 | 70.8 |
| | Winter | 14 | 9.2 | 29.1 | 24.7 | 23 | 7.2 | 36.8 | 18.8 |
| | Spring | 41 | 20.6 | 179.4 | 197.8 | 42 | 20.6 | 184.5 | 82.3 |
| | Summer | 48 | 13 | 212.7 | 275.3 | 49 | 17.2 | 217.1 | 234.9 |
| 2016–2018 | Fall | 121 | 137.2 | 241.9 | 185.3 | 136 | 133.6 | 264.2 | 195.0 (14.5%) |
| | Winter | 86 | 105.6 | 161.9 | 65.6 | 86 | 67.6 | 149.8 | 117.6 |
| | Spring | 118 | 110.6 | 501.3 | 525.7 | 134 | 111 | 546.1 | 445.7 |
| | Summer | 150 | 52.8 | 616.0 | 687.5 | 159 | 746 | 662.7 | 580.3 |
| Total | | 475 | 406.2 | 1521.0 | 1464.2 | 515 | 386.2 | 1622.9 | 1338.6 |

**Table A7.** Cumulative ET and P in mm for the lysimeters in common validated days for reference (lysimeter 1) and non-internal water storage configurations (lysimeters 3 and 4).

| Year | Seasons | Days (1 vs. 3) | P | ET1 | ET3 | Days (1 vs. 4) | P | ET1 | ET4 |
|------|---------|------|------|------|------|------|------|------|------|
| 2016–2017 | Fall | 26 | 47.6 | 64.9 | 59.0 | 26 | 37 | 59.1 | 54 |
| | Winter | 13 | 17.6 | 22.4 | 11.7 | 14 | 17.6 | 25.9 | 13 |
| | Spring | 64 | 87 | 242.0 | 215.5 | 62 | 73.4 | 236.8 | 207.6 |
| | Summer | 38 | 39.2 | 205.7 | 143.6 | 36 | 26.4 | 205.5 | 161.3 |
| 2017–2018 | Fall | 53 | 47.6 | 105.3 | 56.4 | 59 | 48.4 | 113.6 | 46 |
| | Winter | 24 | 30.6 | 73.4 | 34.9 | 21 | 21 | 59.5 | 21.3 |
| | Spring | 31 | 22.6 | 133.7 | 110.6 | 31 | 22.6 | 133.7 | 104.3 |
| | Summer | 72 | 17.6 | 239.9 | 169.5 | 70 | 9.2 | 234.8 | 160.1 |
| 2018–2019 | Fall | 32 | 50 | 75.1 | 44.1 | 30 | 49 | 72.2 | 35.7 |
| | Winter | 21 | 5 | 36.5 | 19.3 | 22 | 5.2 | 36.5 | 12.8 |
| | Spring | 40 | 20.6 | 169.5 | 151.8 | 40 | 20.6 | 169.5 | 124.2 |
| | Summer | 48 | 13 | 212.7 | 117.7 | 47 | 13 | 207.3 | 147.7 |
| 2016–2018 | Fall | 113 | 145.8 | 248.0 | 162.4 | 117 | 135 | 247.5 | 135.9 |
| | Winter | 58 | 53.2 | 132.3 | 65.9 | 57 | 43.8 | 122 | 47.2 |
| | Spring | 135 | 130.2 | 545.2 | 477.9 | 113 | 116.6 | 540 | 436.2 |
| | Summer | 158 | 69.8 | 658.3 | 430.8 | 153 | 48.6 | 647.6 | 469.2 |
| Total | | 464 | 399 | 1583.9 | 1137.0 | 460 | 344 | 1557.1 | 1088.4 |

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
