# Peer review of "Determinants of Evapotranspiration in Urban Rain Gardens: A Case Study with Lysimeters under Temperate Climate"

_hydrology, doi:10.3390/hydrology9030042_

Round 1
Reviewer 1 Report
The article deals with interesting analysis of evapotranspiration in urban area based on measurements in lysimeters. A fully sufficient monitoring data together with the set of parameters from automatic weather station for the period of 3 years (2016-2019) have been considered and overall the results are supported enough by hydrological observations. Even though the topic is not entirely new and such measurements are often carried out, it should be stated that the work has elements developing the achievements to date. In addition, it is clear in perception, well arranged, you can see attention to the details of the research carried out. As far as I know the idea of improving ET measurements in urban areas is of great importance for attempts to model recharge processes and therefore should be treated as somehow innovative approach to this matter.
The presented statistical relationships between the obtained measurements make a convincing impression and correlate with the conclusions. And comparing the measurements with the ET scores according to all the best formulas is a valuable indicator of how to optimize the ET calculations in urban areas.
It would be advisable to extend the description of the research area (section Study area?), with some general basic information about natural conditions, especially considering regional climatic/hydrological data.
The figures are of good quality and legible, however it would be good to insert at least a topographic map in appropriate scale with study area and marked sides of the world, otherwise it is faintly visible in the Fig 1. Fig. B1 (Appendix B) could be considered for insertion directly in the text.
The title should be corrected, without this information about 8 lysimeters, after all, it's not the key issue.
You still need to check the punctuation, in some places, for example, missing spaces (l. 53, l. 75, l.78, tab. 1 - mass(kg); l. 521, l. 531, l. 551) etc.
- 339, 529 – Lysimeters with a lowercase letter
The paper is clearly written, with sufficient explanation of the methods. The problem was solved and the adequate conclusions are presented.
All the references seem to be sufficient and cited in the paper, however the list could be supplemented with an example of the use of ET on the basis of satellite images in groundwater modeling of the catchment e.g.:
Lubczynski M., Gurwin J., 2005: Integration of various data sources for transient groundwater modeling with spatio-temporally variable fluxes—Sardon study case, Spain. Journal of Hydrology 306 (1-4): 71-96.
Check the editions of the list, because, for example, the year of publication is not marked in bold everywhere.
So, please consider the suggestions given above.
Reviewer 2 Report
Evapotranspiration in urban rain gardens: A study of eight lysimeters under temperate climate
Abstract
The need of this research is missing in the abstract. Authors might have to re-think of including that.
Introduction
The research gap is one of the most important sections in a manuscript. This is clearly missing here. The authors need to showcase it. If not there is no point of presenting their experimental work.
In addition, take more recent research work. SUDS are famous in many countries.
Materials and methodology
Table 1: use the SI way to express the decimals.
The authors have conducted a set of experiment, which are appreciated. However, what is the basis of these experiments? Why did you use something else?
What was the reason for this? How do you judge which gives better? And what is the distance from your station to the Météo-France station?
“the rainfall data were compared with the data from Météo-France and Info-climat data and the improbable data were removed.”
Results
Nice periodic views can be seen from Figure 3. What do you expect from these results?
Figure 4 – R2 values are low. What are your facts?
Conclusions
They are based on the work. However, the authors have not presented the research merit here. Why you wanted to try this and what are the take home message from this?
Reference
Better to have recent literature
Round 2
Reviewer 2 Report
Revisions are accepted.